# CLDA: Contrastive Learning for Semi-Supervised Domain Adaptation

**Ankit Singh**
Department of Computer Science
Indian Institute of Technology, Madras
`singh.ankit@cse.iitm.ac.in`

## Abstract

Unsupervised Domain Adaptation (UDA) aims to align the labeled source distribution with the unlabeled target distribution to obtain domain invariant predictive models. However, the application of well-known UDA approaches does not generalize well in Semi-Supervised Domain Adaptation (SSDA) scenarios where few labeled samples from the target domain are available. This paper proposes a simple **C**ontrastive **L**earning framework for semi-supervised **D**omain **A**daptation (**CLDA**) that attempts to bridge the intra-domain gap between the labeled and unlabeled target distributions and the inter-domain gap between source and unlabeled target distribution in SSDA. We suggest employing class-wise contrastive learning to reduce the inter-domain gap and instance-level contrastive alignment between the original(input image) and strongly augmented unlabeled target images to minimize the intra-domain discrepancy. We have empirically shown that both of these modules complement each other to achieve superior performance. Experiments on three well-known domain adaptation benchmark datasets, namely DomainNet, Office-Home, and Office31, demonstrate the effectiveness of our approach. CLDA achieves state-of-the-art results on all the above datasets.

## 1 Introduction

Deep Convolutional networks [30, 52] have shown impressive performance in various computer vision tasks, e.g., image classification [19, 22] and action recognition [48, 23, 57, 32]. However, there is an inherent problem of generalizability with deep-learning models, *i.e.*, models trained on one dataset(source domain) does not perform well on another domain. This loss of generalization is due to the presence of domain shift [11, 55] across the dataset. Recent works [46, 29] have shown that the presence of few labeled data from the target domain can significantly boost the performance of the convolutional neural network(CNN) based models. This observation led to the formulation of Semi-Supervised Domain Adaption (SSDA), which is a variant of Unsupervised Domain Adaptation where we have access to a few labeled samples from the target domain.

Unsupervised domain adaptation methods [42, 12, 36, 51, 35] try to transfer knowledge from the label rich source domain to the unlabeled target domain. Many such existing domain adaptation approaches [42, 12, 51] align the features of the source distribution with the target distribution without considering the category of the samples. These class-agnostic methods fail to generate discriminative features when aligning global distributions. Recently, owing to the success of contrastive approaches [6, 18, 39], in self-representation learning, some recent works [26, 28] have turned to instance-based contrastive approaches to reduce discrepancies across domains.

[46] reveals that the direct application of the well-known UDA approaches in Semi-Supervised Domain Adaptation yields sub-optimal performance. [29] has shown that supervision from labeled source and target samples can only ensure the partial cross-domain feature alignment. This creates

35th Conference on Neural Information Processing Systems (NeurIPS 2021).

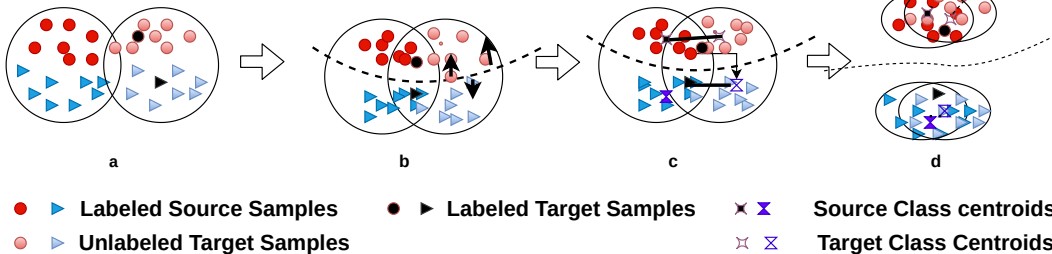

| ● ▶ **Labeled Source Samples** | ● ▶ **Labeled Target Samples** | ✕ ✕ **Source Class centroids** |
| ○ ▶ **Unlabeled Target Samples** | | ⋈ ✕ **Target Class Centroids** |

Figure 1: **Conceptual description of CLDA approach**. **(a)** Intial distribution of samples from both domain .**(b)** Instance Contrastive Alignment ensures unlabeled target samples move into the low entropy area forming robust clusters **(c)** Inter-Domain Contrastive Alignment minimizes the distance between the clusters of same class from both domain **(d)** The clusters of both domain are well aligned and samples are far away from decision boundary.

aligned and unaligned sub-distributions of the target domain, causing intra-domain discrepancy apart from inter-domain discrepancy in SSDA.

In this work, we propose CLDA, a simple single-stage novel contrastive learning framework to address the aforementioned problem. Our framework contains two significant components to learn domain agnostic representation. First, Inter-Domain Contrastive Alignment reduces the discrepancy between centroids of the same class from the source and the target domain while increasing the distance between the class centroids of different classes from both source and target domain. This ensures clusters of the same class from both domains are near each other in latent space than the clusters of the other classes from both domains.

Second, inspired by the success of self-representation learning in semi-supervised settings [17, 6, 49], we propose to use Instance Contrastive Alignment to reduce the intra-domain discrepancy. In this, we first generate the augmented views of the unlabeled target images using image augmentation methods. Alignment of the features of the original and augmented images of the unlabeled samples from the target domain ensures that they are closer to each other in latent space. The alignment between two variants of the same image ensures that the classifier boundary lies in the low-density regions assuring that the feature representations of two variants of the unlabeled target images are similar, which helps to generate better clusters for the target domain.

In summary, our key contributions are as follows. 1) We propose a novel, simple single-stage training framework for Semi-supervised Domain Adaptation. 2)We propose using alignment at class centroids and instance levels to reduce inter and intra domain discrepancies present in SSDA. 3)We evaluate the effectiveness of different augmentation approaches, for instance-based contrastive alignment in the SSDA setting. 4)We evaluate our approach over three well-known Domain Adaptation datasets (DomainNet, Office-Home, and Office31) to gain insights. Our approach achieves the state of the art results across multiple datasets showing its effectiveness. We perform extensive ablation experiments highlighting the role of different components of our framework.

## 2   Related Works

### 2.1   Unsupervised Domain Adaptation

Unsupervised Domain Adaptation (UDA) [14] is a well-studied problem, and most UDA algorithms reduce the domain gap by matching the features of the sources and target domain [16, 4, 24, 36, 51, 27]. Feature-based alignment methods reduce the global divergence [16, 51] between source and target distribution. Adversarial learning [12, 5, 34, 35, 42, 41] based approaches have shown impressive performance in reducing the divergence between source and target domains. It involves training the model to generate features to deceive the domain classifier, invariantly making the generated features domain agnostic. Recently, Image translation methods [20, 21, 38] have been explored in UDA where an image from the target domain is translated to the source domain to be treated as an image from the source domain to overcome the divergence present across domains.

Despite remarkable progress in UDA, [46] shows the UDA approaches do not perform well in the SSDA setting, which we consider in this work.

## 2.2 Semi-Supervised Learning

Semi-Supervised Learning(SSL) aims to leverage the vast amount of unlabeled data with limited labeled data to improve classifier performance. The main difference between SSL and SSDA is that SSL uses data sampled from the same distribution while SSDA deals with data sampled from two domains with inherent domain discrepancy. The current line of work in SSL [50, 3, 31, 10] follows consistency-based approaches to reduce the intra-domain gap. Mean teacher [53] uses two copies of the same model (student model and teacher model) to ensure consistency across augmented views of the images. Weights of the teacher model are updated as the exponential moving average of the weights of the student model. Mix-Match [3] and ReMixMatch [2] use interpolation between labeled and unlabeled data to generate perturbed features. Recently introduced FixMatch [50] achieves impressive performance using the confident pseudo labels of the unlabeled samples and treating them as labels for the strongly perturbed samples. However, direct application of SSL in the SSDA setting yields sub-optimal performance as the presumption in the SSL is that distributions of labeled and unlabeled data are identical, which is not the case in SSDA.

## 2.3 Contrastive Learning

Contrastive Learning(CL) has shown impressive performance in self-representation learning [6, 1, 18, 54, 39]. Most contrastive learning methods align the representations of the positive pair (similar images) to be close to each other while making negative pairs apart. In semantic segmentation, [33] uses patch-wise contrastive learning to reduce the domain divergence by aligning the similar patches across domains. In domain adaptation, contrastive learning [28, 26] has been applied for alignment at the instance level to learn domain agnostic representations. [26, 28] use samples from the same class as positive pairs, and samples from different classes are counted as negative pairs. [26] modifies Maximum Mean Discrepancy (MMD) [16] loss to be used as a contrastive loss. In contrast to [28, 26], our work proposes to use contrastive learning in SSDA setting both at the class and instance level (across perturbed samples of the same image) to learn the semantic structure of the data better.

## 2.4 Semi-Supervised Domain Adaptation

Semi-Supervised Domain Adaptation (SSDA) aims to reduce the discrepancy between the source and target distribution in the presence of limited labeled target samples. [46] first proposed to align the source and target distributions using adversarial training. [29] shows the presence of intra domain discrepancy in the target distribution and introduces a framework to mitigate it. [25] uses consistency alongside multiple adversarial strategies on top of MME [46]. [9] introduced the meta-learning framework for Semi-Supervised Domain Adaptation. [58] breaks down the SSDA problem into two subproblems, namely, SSL in the target domain and UDA problem across the source and target domains, and learn the optimal weights of the network using co-training. [37] proposed to use pretraining of the feature extractor and consistency across perturbed samples as a simple yet effective strategy for SSDA. [44] introduces a framework for SSDA consisting of a shared feature extractor and two classifiers with opposite purposes, which are trained in an alternative fashion; where one classifier tries to cluster the target samples while the other scatter the source samples, so that target features are well aligned with source domain features. Most of the above approaches are based on adversarial training, while our work proposes to use contrastive learning-based feature alignment at the class level and the instance level to reduce discrepancy across domains.

# 3 Methodology

In this section, we present our novel Semi-Supervised Domain Adaptation approach to learn domain agnostic representation. We will first introduce the background and notations used in our work and then describe our approach and its components in detail.

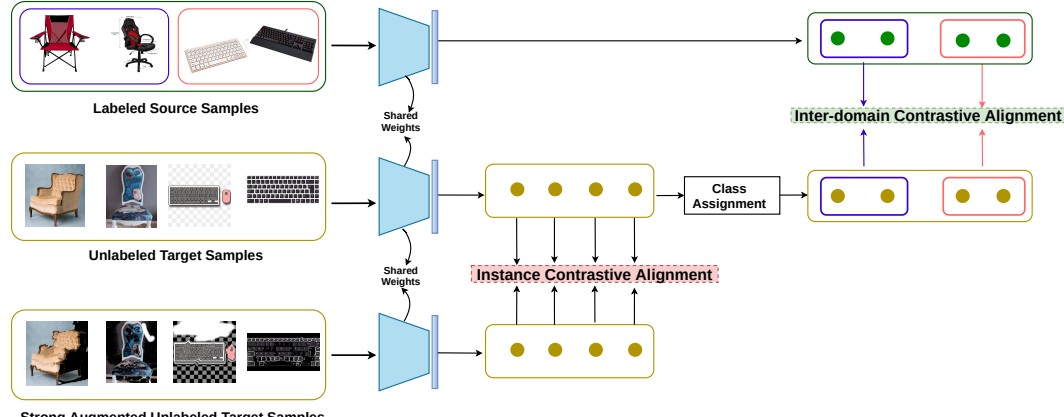

Figure 2: **Outline of our CLDA Framework** Our approach consists of aligning the outputs of the neural network at two levels. At the instance level, we try to maximize the similarity between features of unlabeled target images and strongly augmented unlabeled target images using Instance Contrastive Alignment. At the class level, we pass the images from both domains through the network, where we assign the labels to features of unlabeled target images and compute the centroids of each class of the target domain. Similarly, we compute the centroids for source domain features using their class labels. Finally, we maximize the similarity between centroids of the same class across domains by employing Inter-Domain Contrastive Alignment. We also used cross-entropy loss on the labeled source and target images, apart from the above components in our framework.

## 3.1 Problem Formulation

In Semi-Supervised Domain Adaptation, we have datasets sampled from two domains. The source dataset contains labeled images $\mathcal{D}_s = \{(x_i^s, y_i^s)\}_{i=1}^{N_s} \subset \mathcal{R}^d \times \mathcal{Y}$ sampled from some distribution $P_S(X, Y)$. Besides that, we have two sets of data sampled from target domain distribution $P_T(X, Y)$. We denote the labeled set of images sampled from the target domain as $\mathcal{D}_{lt} = \{(x_i^{lt}, y_i^{lt})\}_{i=1}^{N_{lt}}$. The unlabeled set sampled from target domain $\mathcal{D}_t = \{(x_i^t)\}_{i=1}^{N_t}$ contains large number of images $(N_t \gg N_{lt})$ without any corresponding labels associated with them. We also denote the labeled data from both domains as $\mathcal{D}_l = \mathcal{D}_s \cup \mathcal{D}_{lt}$. Labels $y_i^s$ and $y_i^{lt}$ of the samples from source and labeled target set correspond to one of the categories of the dataset having $K$ different classes/categories *i.e* $Y = \{1, 2, ...K\}$. Our goal is to learn a task specific classifier using $D_s, D_{lt}$ and $D_t$ to accurately predict labels on test data from target domain.

## 3.2 Supervised Training

Labeled source and target samples are passed through the CNN-based feature extractor $\mathcal{G}(.)$ to obtain corresponding features, which are then passed through task-specific classifier $\mathcal{F}(.)$ to minimize the well-known cross-entropy loss on the labeled images from both source and target domains.

$$\mathcal{L}_{sup} = -\sum_{k=1}^{K} (y^i)_k \log(\mathcal{F}(\mathcal{G}((x_l^i))_k \tag{1}$$

## 3.3 Inter-Domain Contrastive Alignment

Our method is based on the observation that the samples from the same category across domains must cluster in the latent space. However, this is observed only for the source domain due to the availability of the labels. Samples from the target domain do not align to form clusters due to the domain shift between the target and the source distributions. This discrepancy between the cluster of the same category across domains is reduced by aligning the centroids of each class of source and target domain. [6, 17] have shown that having a separate projection space is beneficial for contrastive

training. Instead of using a separate projection, we have used the outputs from the task-specific classifier as features to align the clusters across the domain.

We represent the centroid of the images from the source domain belonging to class $k$ as the mean of their features, which can be written as

$$C_k^s = \frac{\sum\limits_{i=1}^{i=B} \mathbb{1}_{\{y_i^s=k\}} \mathcal{F}(\mathcal{G}(x_i^s))}{\sum\limits_{i=1}^{i=B} \mathbb{1}_{\{y_i^s=k\}}} \tag{2}$$

where $B$ is the size of batch. We maintain a memory bank $\left(C^s = [C_1^s, C_2^s, ....C_K^s]\right)$ to store the centroids of each class from source domain. We use exponential moving average to update these centroid values during the training

$$C_k^s = \rho(C_k^s)_{step} + (1 - \rho)(C_k^s)_{step-1}$$

where $\rho$ is a momentum term, and $(C_k^s)_{step}$ and $(C_k^s)_{step-1}$ are the centroid values of class $k$ at the current and previous step, respectively.

We also need to cluster the unlabeled target samples for Inter-Domain Contrastive Alignment. The pseudo labels obtained from the task specific classifier as shown in Eq (3) is used as the class labels for the corresponding unlabeled target samples.

$$\hat{y}_i^t = argmax((\mathcal{F}(\mathcal{G}(x_i^t)))) \tag{3}$$

Similar to the source domain , we also calculate the separate cluster centroid $C_k^t$ for each of the class $k$ of the target samples present in the minibatch as per the Eq (2) where unlabeled target images replace the images from the source domain with their corresponding pseudo label. The model is then trained to maximize the similarity between the cluster representation of each class $k$ from the source and the target domain. $C_k^s$ and $C_k^t$ form the positive pair while the remaining cluster centroids from both domains form the negative pairs. The remaining clusters from both domains are pushed apart in the latent space. This is achieved through employing a modified NT-Xent (normalized temperature-scaled cross-entropy) contrastive loss [6, 39, 49, 33] for domain adaptation given by

$$\mathcal{L}_{clu}(C_i^t, C_i^s) = -\log \frac{h(C_i^t, C_i^s)}{h(C_i^t, C_i^s) + \sum\limits_{\substack{r=1 \\ q\in\{s,t\}}}^{K} \mathbb{1}_{\{r\neq i\}} h(C_i^t, C_r^q)} \tag{4}$$

where $h(\mathbf{u}, \mathbf{v}) = \exp\left(\frac{\mathbf{u}^\top \mathbf{v}}{||\mathbf{u}||_2 ||\mathbf{v}||_2}/\tau\right)$ measures the exponential of cosine similarity , $\mathbb{1}$ is an indicator function and $\tau$ is the temperature hyperparameter.

### 3.4 Instance Contrastive Alignment

Recent works on contrastive learning [18, 39, 6] show encouraging results in single domain settings. [28] extends contrastive learning into multi-domain settings. Inspired by such success, we employ Instance Contrastive Learning to form stable and correct cluster cores in the target domain.

To perform contrastive alignment at the instance level, we first generate a strongly augmented version of the unlabeled target image *i.e* $\tilde{x}_i^t = \psi(x_i^t)$ where $\psi(.)$ is the strong augmentation function [8]. Next, we employ the NT-Xent loss [6, 39] as defined in Eq (5) to ensure that these two variants of the same image are closer to each other in the latent space while the rest of the images in minibatch of size $B$ are pushed apart. This idea stems from the cluster assumption in an ideal classifier, which states the decision boundary should lie in the low-density region, ensuring consistent prediction for different augmented variants of the same image.

$$\mathcal{L}_{ins}(\tilde{x}_i^t, x_i^t) = -\log \frac{h(\mathcal{F}(\mathcal{G}(\tilde{x}_i^t), \mathcal{F}(\mathcal{G}(x_i^t)))}{\sum\limits_{r=1}^{B} h(\mathcal{F}(\mathcal{G}(\tilde{x}_i^t)), \mathcal{F}(\mathcal{G}(x_r^t))) + \sum\limits_{r=1}^{B} \mathbb{1}_{\{r\neq i\}} h(\mathcal{F}(\mathcal{G}(\tilde{x}_i^t)), \mathcal{F}(\mathcal{G}(\tilde{x}_r^t)))} \tag{5}$$

In SSDA, [29] has shown that target distribution gets divided into aligned and unaligned subdistribution in the presence of very few labeled target data. Thus, aligning the unaligned subdistribution can lead to improved performance, while perturbing the aligned sub-distribution can result in a negative transfer. Therefore, we only propagate the gradients for strongly augmented images to avoid perturbing the aligned sub-distribution in the target domain.

[6] shows stronger augmentation in contrastive learning leads to improved performance. Consistent prediction across the input and strongly augmented unlabeled images in Instance Contrastive Alignment forces the unaligned target sub-distribution to move away from the low-density region towards aligned distribution. This ensures better clustering in the unlabeled target distribution, which is validated by improved accuracy as shown in Table 5 after employing Instance Contrastive Alignment with Inter-Domain Contrastive Alignment.

Both of the components of the CLDA framework are necessary for the improved performance, as shown in Table 5 . Instance Contrastive Alignment ensures that unlabeled target samples are consistent and are in the high-density region. However, it does not assure alignment between source and unlabeled target samples. Inter-Domain Contrastive Alignment reduces the discrepancy between unlabeled target samples and source domain but unlabeled target samples closer to the decision boundary might get pushed towards the wrong classes resulting in negative transfer. Thus, combining both components results in a much better alignment of the unlabeled target samples towards the source domain, leading to improved performance of the framework.

### 3.5 Overall framework and training objective

The overall training objective employs supervised loss, Inter-Domain Contrastive Alignment and Instance Contrastive Alignment which can be formulated as follows:

$$\mathcal{L}_{tot} = \mathcal{L}_{sup} + \alpha * \mathcal{L}_{clu} + \beta * \mathcal{L}_{ins} \tag{6}$$

We train the model in our framework by employing overall training loss described as in (6).

## 4 Experiments

### 4.1 Experimental Setup

We evaluate the effectiveness of our approach on three different domain adaptation datasets: DomainNet [43], Office-Home [56] and Office31 [45]. DomainNet [43] is a large-scale domain adaptation dataset with 345 classes across 6 domains. Following MME [46], we use a subset of the dataset containing 126 categories across four domains: Real(R), Clipart(C), Sketch(S), and Painting(P). The performance on DomainNet is evaluated using 7 different combinations out of possible 12 combinations. Office-Home [56] is another widely used domain adaptation benchmark dataset with 65 classes across four domains: Art(Ar), Product(Pr), Clipart(Cl), and Real (Rl). We perform experiments on all possible combinations of 4 domains. Office31 [45] is a relatively smaller dataset containing just 31 categories of data across three domains- Amazon(A), Dslr(D), Webcam(W). Following prior work [46, 29], we evaluate our approach on two combinations for the office31 dataset.

For the fair comparison, we use the data-splits (train, validation, and test splits) released by [46] on Github [1]. We use the same settings for the benchmark datasets as in the prior work [46, 29], including the number of labeled samples in the target domain, which are consistent across all experiments.

### 4.2 Implementation Details

Similar to the previous works on SSDA [46, 29, 9], we use Resnet34 and Alexnet as the backbone networks in our paper. We only used VGG for Office31 due to its higher memory requirements. The feature generator model is initialized with ImageNet weights, and the classifier is randomly initialized and has the same architecture as in [46, 29, 9]. All our experiments are performed using Pytorch [40]. We use an identical set of hyperparameters ($\alpha = 4$, $\beta = 1$ ) across all our experiments other

---

[1] https://github.com/VisionLearningGroup/SSDA_MME

| Net | Method | Rl→Cl | Rl→Pr | Rl→Ar | Pr→Rl | Pr→Cl | Pr→Ar | Ar→Pl | Ar→Cl | Ar→Rl | Cl→Rl | Cl→Ar | Cl→Pr | Mean |
|---|---|---|---|---|---|---|---|---|---|---|---|---|---|---|
| Alexnet | S+T | 44.6 | 66.7 | 47.7 | 57.8 | 44.4 | 36.1 | 57.6 | 38.8 | 57.0 | 54.3 | 37.5 | 57.9 | 50.0 |
| | DANN | 47.2 | 66.7 | 46.6 | 58.1 | 44.4 | 36.1 | 57.2 | 39.8 | 56.6 | 54.3 | 38.6 | 57.9 | 50.3 |
| | ADR | 37.8 | 63.5 | 45.4 | 53.5 | 32.5 | 32.2 | 49.5 | 31.8 | 53.4 | 49.7 | 34.2 | 50.4 | 44.5 |
| | CDAN | 36.1 | 62.3 | 42.2 | 52.7 | 28.0 | 27.8 | 48.7 | 28.0 | 51.3 | 41.0 | 26.8 | 49.9 | 41.2 |
| | ENT | 44.9 | 70.4 | 47.1 | 60.3 | 41.2 | 34.6 | 60.7 | 37.8 | 60.5 | 58.0 | 31.8 | 63.4 | 50.9 |
| | MME | 51.2 | 73.0 | 50.3 | 61.6 | 47.2 | 40.7 | 63.9 | 43.8 | 61.4 | 59.9 | 44.7 | 64.7 | 55.2 |
| | Meta-MME | 50.3 | - | - | - | 48.3 | 40.3 | - | 44.5 | - | - | 44.5 | - | - |
| | BiAT | - | - | - | - | - | - | - | - | - | - | - | - | 56.4 |
| | APE | **51.9** | **74.6** | 51.2 | 61.6 | 47.9 | 42.1 | 65.5 | 44.5 | 60.9 | 58.1 | 44.3 | 64.8 | 55.6 |
| | **CLDA**(ours) | 51.5 | 74.1 | **54.3** | **67.0** | **47.9** | **47.0** | **65.8** | **47.4** | **66.6** | **64.1** | **46.8** | **67.5** | **58.3** |
| Resnet34 | S+T | 55.7 | 80.8 | 67.8 | 73.1 | 53.8 | 63.5 | 73.1 | 54.0 | 74.2 | 68.3 | 57.6 | 72.3 | 66.2 |
| | DANN | 57.3 | 75.5 | 65.2 | 69.2 | 51.8 | 56.6 | 68.3 | 54.7 | 73.8 | 67.1 | 55.1 | 67.5 | 63.5 |
| | ENT | 62.6 | 85.7 | 70.2 | 79.9 | 60.5 | 63.9 | 79.5 | 61.3 | 79.1 | 76.4 | 64.7 | 79.1 | 71.9 |
| | MME | 64.6 | 85.5 | 71.3 | 80.1 | 64.6 | 65.5 | 79.0 | 63.6 | 79.7 | 76.6 | 67.2 | 79.3 | 73.1 |
| | Meta-MME | 65.2 | - | - | - | 64.5 | 66.7 | - | 63.3 | - | - | 67.5 | - | - |
| | APE | **66.4** | 86.2 | 73.4 | 82.0 | **65.2** | 66.1 | 81.1 | **63.9** | 80.2 | 76.8 | 66.6 | 79.9 | 74.0 |
| | **CLDA** (ours) | 66.0 | **87.6** | **76.7** | **82.2** | 63.9 | **72.4** | **81.4** | 63.4 | **81.3** | **80.3** | **70.5** | **80.9** | **75.5** |

Table 1: **Performance Comparison in Office-Home.** Numbers show top-1 accuracy values for different domain adaptation scenarios under 3-shot setting using Alexnet and Resnet34 as backbone networks. We have highlighted the best method for each transfer task. CLDA surpasses all the baseline methods in most adaptation scenarios. Our Proposed framework achieves the best average performance among all compared methods.

than minibatch size. All the hyperparameters values are decided using validation performance on Product to Art experiments on the Office-Home dataset. We have set $\tau = 5$ in our experiments. Each minibatch of size $B$ contains an equal number of source and labeled target examples, while the number of unlabeled target samples is $\mu \times B$. We study the effect of $\mu$ in section 4.5. Resnet34 experiments are performed with minibatch size, $B = 32$ and Alexnet models are trained with $B = 24$. We use $\mu = 4$ for all our experiments. We use SGD optimizer with a momentum of $0.9$ and an initial learning rate of $0.01$ with cosine learning rate decay for all our experiments. Weight decay is set to $0.0005$ for all our models. Other details of the experiments are included in the supplementary.

### 4.3 Baselines

We compare our CLDA framework with previous state-of-the-art SSDA approaches : **MME** [46], **APE** [29], **BiAT** [25] , **UODA** [44], **Meta-MME** [9] and **ENT** [15] using the performance reported by these papers. papers. We also included the results from adversarial based baseline methods: **DANN** [13], **ADR** [47] and **CDAN** [35] as reported in [46]. We also provide the **S+T** results where the model is trained using all the labeled samples across domains.

### 4.4 Results

Table 1- 3 show top-1 accuracies and mean accuracies for different combination of domain adaptation scenarios for all three datasets in comparison with baseline SSDA methods.

**Office-Home.** Table 1 contains the results of the Office-Home dataset for 3-shot setting with Alexnet and Resnet34 as backbone networks. Results for the 1-shot adaptation scenarios are included in the supplementary. Our method consistently performs better than the baseline approaches and achieves $58.3\%$ and $75.5\%$ mean accuracy with Alexnet and Resnet34, respectively. Our approach surpasses the state-of-the-art SSDA approaches in most of the adaptation tasks. In some domain adaptation cases, such as Cl → Rl, Rl → Ar and Pr → Ar, we exceeded APE by more than $3\%$.

**DomainNet**: Our CLDA approach surpasses the performance of existing SSDA baselines as shown in Table 2. Using Alexnet backbone, our method improves over BiAT by $5.2\%$ and $4.9\%$ in 1-shot and 3-shot settings, respectively. We obtain similarly improved performance when we switch the neural backbone from Alexnet to Resnet34. With Resnet34 as the backbone, we gain $4.3\%$ and $3.6\%$ over APE in 1-shot and 3-shot settings, respectively. Similar to the Office-Home, our approach surpasses the well-known domain adaptation benchmarks methods in most domain adaptation tasks of the DomainNet dataset. Such consistent improved performance shows that our approach reduces both inter and intra domain discrepancy prevalent in SSDA.

| Net | Method | R→C | | R→P | | P→C | | C→S | | S→P | | R→S | | P→R | | Mean | |
|---|---|---|---|---|---|---|---|---|---|---|---|---|---|---|---|---|---|
| | | 1-shot | 3-shot | 1-shot | 3-shot | 1-shot | 3-shot | 1-shot | 3-shot | 1-shot | 3-shot | 1-shot | 3-shot | 1-shot | 3-shot | 1-shot | 3-shot |
| Alexnet | S+T | 43.3 | 47.1 | 42.4 | 45.0 | 40.1 | 44.9 | 33.6 | 36.4 | 35.7 | 38.4 | 29.1 | 33.3 | 55.8 | 58.7 | 40.0 | 43.4 |
| | DANN | 43.3 | 46.1 | 41.6 | 43.8 | 39.1 | 41.0 | 35.9 | 36.5 | 36.9 | 38.9 | 32.5 | 33.4 | 53.5 | 57.3 | 40.4 | 42.4 |
| | ADR | 43.1 | 46.2 | 41.4 | 44.4 | 39.3 | 43.6 | 32.8 | 36.4 | 33.1 | 38.9 | 29.1 | 32.4 | 55.9 | 57.3 | 39.2 | 42.7 |
| | CDAN | 46.3 | 46.8 | 45.7 | 45.0 | 38.3 | 42.3 | 27.5 | 29.5 | 30.2 | 33.7 | 28.8 | 31.3 | 56.7 | 58.7 | 39.1 | 41.0 |
| | ENT | 37.0 | 45.5 | 35.6 | 42.6 | 26.8 | 40.4 | 18.9 | 31.1 | 15.1 | 29.6 | 18.0 | 29.6 | 52.2 | 60.0 | 29.1 | 39.8 |
| | MME | 48.9 | 55.6 | 48.0 | 49.0 | 46.7 | 51.7 | 36.3 | 39.4 | 39.4 | 43.0 | 33.3 | 37.9 | 56.8 | 60.7 | 44.2 | 48.2 |
| | Meta-MME | - | 56.4 | - | 50.2 | | 51.9 | - | 39.6 | - | 43.7 | - | 38.7 | - | 60.7 | - | 48.8 |
| | BiAT | 54.2 | 58.6 | 49.2 | 50.6 | 44.0 | 52.0 | 37.7 | 41.9 | 39.6 | 42.1 | 37.2 | 42.0 | 56.9 | 58.8 | 45.5 | 49.4 |
| | APE | 47.7 | 54.6 | 49.0 | 50.5 | 46.9 | 52.1 | 38.5 | 42.6 | 38.5 | 42.2 | 33.8 | 38.7 | 57.5 | 61.4 | 44.6 | 48.9 |
| | **CLDA** (ours) | **56.3** | **59.9** | **56.0** | **57.2** | **50.8** | **54.6** | **42.5** | **47.3** | **46.8** | **51.4** | **38.0** | **42.7** | **64.4** | **67.0** | **50.7** | **54.3** |
| Resnet34 | S+T | 55.6 | 60.0 | 60.6 | 62.2 | 56.8 | 59.4 | 50.8 | 55.0 | 56.0 | 59.5 | 46.3 | 50.1 | 71.8 | 73.9 | 56.9 | 60.0 |
| | DANN | 58.2 | 59.8 | 61.4 | 62.8 | 56.3 | 59.6 | 52.8 | 55.4 | 57.4 | 59.9 | 52.2 | 54.9 | 70.3 | 72.2 | 58.4 | 60.7 |
| | ADR | 57.1 | 60.7 | 61.3 | 61.9 | 57.0 | 60.7 | 51.0 | 54.4 | 56.0 | 59.9 | 49.0 | 51.1 | 72.0 | 74.2 | 57.6 | 60.4 |
| | CDAN | 65.0 | 69.0 | 64.9 | 67.3 | 63.7 | 68.4 | 53.1 | 57.8 | 63.4 | 65.3 | 54.5 | 59.0 | 73.2 | 78.5 | 62.5 | 66.5 |
| | ENT | 65.2 | 71.0 | 65.9 | 69.2 | 65.4 | 71.1 | 54.6 | 60.0 | 59.7 | 62.1 | 52.1 | 61.1 | 75.0 | 78.6 | 62.6 | 67.6 |
| | MME | 70.0 | 72.2 | 67.7 | 69.7 | 69.0 | 71.7 | 56.3 | 61.8 | 64.8 | 66.8 | 61.0 | 61.9 | 76.1 | 78.5 | 66.4 | 68.9 |
| | UODA | 72.7 | 75.4 | 70.3 | 71.5 | 69.8 | 73.2 | 60.5 | 64.1 | 66.4 | 69.4 | 62.7 | 64.2 | 77.3 | 80.8 | 68.5 | 71.2 |
| | Meta-MME | - | 73.5 | - | 70.3 | - | 72.8 | - | 62.8 | - | 68.0 | - | 63.8 | - | 79.2 | - | 70.1 |
| | BiAT | 73.0 | 74.9 | 68.0 | 68.8 | 71.6 | 74.6 | 57.9 | 61.5 | 63.9 | 67.5 | 58.5 | 62.1 | 77.0 | 78.6 | 67.1 | 69.7 |
| | APE | 70.4 | 76.6 | 70.8 | 72.1 | **72.9** | **76.7** | 56.7 | 63.1 | 64.5 | 66.1 | 63.0 | 67.8 | 76.6 | 79.4 | 67.6 | 71.7 |
| | **CLDA** (ours) | **76.1** | **77.7** | **75.1** | **75.7** | 71.0 | 76.4 | **63.7** | **69.7** | **70.2** | **73.7** | **67.1** | **71.1** | **80.1** | **82.9** | **71.9** | **75.3** |

Table 2: **Performance Comparison in DomainNet.** Numbers show Top-1 accuracy values for different domain adaptation scenarios under 1-shot and 3-shot settings using Alexnet and Resnet34 as backbone networks. CLDA achieves better performance than all the baseline methods in most of the domain adaptation tasks. We have highlighted the best approach for each domain adaptation task. Our Proposed framework achieves the best average performance among all compared methods.

| | Alexnet | | | | | | VGG | | | | | |
|---|---|---|---|---|---|---|---|---|---|---|---|---|
| | W→A | | D→A | | Mean | | W→A | | D→A | | Mean | |
| Method | 1-shot | 3-shot | 1-shot | 3-shot | 1-shot | 3-shot | 1-shot | 3-shot | 1-shot | 3-shot | 1-shot | 3-shot |
| S+T | 50.4 | 61.2 | 50.0 | 62.4 | 50.2 | 61.8 | 169.2 | 73.2 | 68.2 | 73.3 | 68.7 | 73.25 |
| DANN | 57.0 | 64.4 | 54.5 | 65.2 | 55.8 | 64.8 | 69.3 | 75.4 | 70.4 | 74.6 | 69.85 | 75.0 |
| ADR | 50.2 | 61.2 | 50.9 | 61.4 | 50.6 | 61.3 | 69.7 | 73.3 | 69.2 | 74.1 | 69.45 | 73.7 |
| CDAN | 50.4 | 60.3 | 48.5 | 61.4 | 49.5 | 60.8 | 65.9 | 74.4 | 64.4 | 71.4 | 65.15 | 72.9 |
| ENT | 50.7 | 64.0 | 50.0 | 66.2 | 50.4 | 65.1 | 69.1 | 75.4 | 72.1 | 75.1 | 70.6 | 75.25 |
| MME | 57.2 | 67.3 | 55.8 | 67.8 | 56.5 | 67.6 | 73.1 | 76.3 | 73.6 | **77.6** | 73.35 | 76.95 |
| BiAT | 57.9 | 68.2 | 54.6 | 68.5 | 56.3 | 68.4 | - | - | - | - | - | - |
| APE | - | 67.6 | - | 69.0 | - | 68.3 | - | - | - | - | - | - |
| CLDA | **64.6** | **70.5** | **62.7** | **72.5** | **63.6** | **71.5** | **76.2** | **78.6** | **75.1** | 76.7 | **75.6** | **77.6** |

Table 3: **Performance Comparison in Office31.** Numbers show Top-1 accuracy values for different domain adaptation scenarios under 1-shot and 3-shot settings using Alexnet and VGG as backbone networks. CLDA outperforms all the baseline approaches in both scenarios. We have highlighted the superior method on each domain adaptation task. Our Proposed framework achieves the best mean accuracy among all baseline methods.

**Office31**: Similar to other datasets, our proposed method with Alexnet and VGG as neural backbone achieves the best performance in both domain adaption scenarios for office31 as shown in Table 3. Using Alexnet backbone, we beat the APE [29] by 3.2% in 3-shot and BiAT by 7.3% in 1-shot settings. We observe similar gains over all the baselines methods with VGG as the neural network backbone. This shows the efficacy of our proposed approach irrespective of the used backbone.

## 4.5 Ablation Studies

We perform extensive ablation experiments to analyze our CLDA framework and the effects of the different components and hyperparameters. We perform these experiments on the 3-shot Pr → Ar domain adaptation task of the Office-Home dataset using Resnet34 unless specified otherwise.

**Effectiveness of Individual Modules**: Our CLDA framework is composed of two modules: Inter-Domain Contrastive Alignment and Instance Contrastive Alignment. We investigate the significance of each component of our framework by dropping the other during training. We observe that the test

| Augmentation | Test Accuracy( Pr → Ar) | Test Accuracy(Rl → Ar) |
|---|---|---|
| Horizontal Flipping (Hflip) | 68.1 | 73.4 |
| Hflip + Color Jitter | 67.6 | 74.9 |
| Hflip+ Color Jitter + Grayscale | 70.2 | 76.2 |
| Rand Augment (RA) [8] | 71.1 | 74.6 |
| RA + Grayscale | 72.4 | 76.7 |
| Auto Augment [7] | 69.9 | 75.3 |

Table 4: **Effect of Strong Augmentations** Numbers show the test accuracy on 3-shot domain adaptation tasks of the Office-Home dataset with Resnet34 with different augmentation policies.

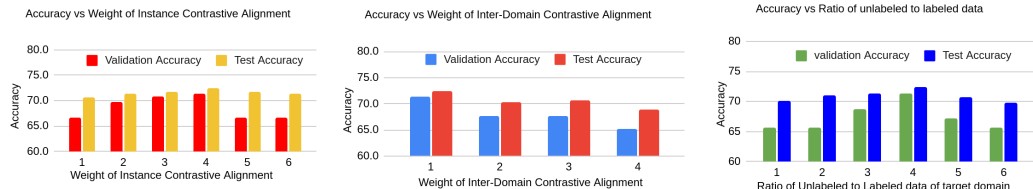

Figure 3: **Effect of different hyperparameters on 3-shot Pr → Ar (Product to Art) data adaptation scenario on the Office-Home using Resnet34. (a)** Effect of varying the weight of Instance Contrastive Alignment on validation and test Accuracy **(b)** Effect of varying weight of Inter-Domain Contrastive Alignment on validation and test Accuracy **(c)** Effect of $\mu$, ratio of unlabeled target to labeled target data on validation and test accuracy.

accuracy drops from $72.4\%$ to $68.3\%$ when only Inter-Domain Contrastive Alignment is used, and it drops to $67.7\%$ when Instance Contrastive Alignment is used alone as shown in Table 5(a). Though individual modules do not yield high performance on their own but once combined, they surpass their individual performance by a margin of around $4\%$.

**Effect of Different Hyperparameters**: We analyze the importance of different hyperparameters used in our approach. We observe that the weight of Instance Contrastive Alignment affects the performance of our approach as the test accuracy drops from $72.4\%$ to $70.7\%$ when we set $\alpha$ to $1$ instead of its optimal value of $4$ as shown in figure 3. We also notice that increasing $\beta$ led to a reduction of the validation and test performance. We also look into the effect of $\mu$, which is the ratio of unlabeled to labeled data in a minibatch. We observe that an increasing value of $\mu$ increases the performance till $\mu = 4$, after which it starts to drop, as shown in figure 3.

**Importance of Instance Contrastive Alignment**: Instance Contrastive Alignment ensures similar representation across different variants of the unlabeled target images. This consistency is also ensured by other well-known SSL approaches like FixMatch [50]. We perform an ablation experiment replacing Instance Contrastive Alignment with FixMatch. We also compare with L1 and L2 loss to have a fair analysis. As shown in Table 5 (b) Instance Contrastive Alignment helps to achieve superior performance in comparison with other consistency-based approaches.

| Approach | Test Accuracy |
|---|---|
| CLDA w/o Instance Contrastive | 68.3 |
| CLDA w/o Inter-Domain Contrastive | 67.7 |
| CLDA (ours) | 72.4 |

(a) Ablation Study on the effectiveness of Individual components of the CLDA framework on Pr → Ar adaptation task of the Office-Home dataset using Resnet34.

| Approach | Test Accuracy |
|---|---|
| Fix-Match | 70.8 |
| L1 loss | 69.4 |
| L2 loss | 69.3 |
| CLDA (ours) | 72.4 |

(b) Ablation Study on other consistency based approaches on Pr → Ar domain adaptation task of the Office-Home using Resnet34.

Table 5: Experiments to understand the significance of individual components of our framework.

| Experiments | 0 samples mislabeled | 8 samples mislabeled ($\sim 12\%$) | 16 samples mislabeled ($\sim 25\%$) |
|---|---|---|---|
| Pr $\to$ Ar | 66.2 | 66.0 | 65.7 |
| Rl $\to$ Ar | 72.6 | 72.05 | 71.56 |

Table 6: **Ablation study to understand the effect of outliers in target domain.** Numbers show the test accuracy of 1-shot domain adaptation tasks of the Office-Home dataset with Resnet34.

**Effect of Other Clustering Techniques**: Inter-Domain Contrastive Alignment requires pseudo labels for the unlabeled target data for clustering. In this ablation experiment, we replace our approach of using the model's prediction as a pseudo label with K-means clustering, which we invoke after every 50 steps and use the generated centroids for the next 50 steps to obtain pseudo-class labels for unlabeled target data. We observe a drop in performance (from $72.4\%$ to $71.2\%$) when using K-means to obtain the pseudo label for unlabeled target images.

**Effect of Augmentation Policy**: We look into different augmentation policies for the Instance Contrastive Alignment. As suggested in [6], a stronger augmentation policy for contrastive learning increases the performance of the model. We find that RandAugment [8] with Grayscale augmentation policy gives better results over other augmentation policies. The influence of the strong augmentation can be observed from $\sim 4\%$ improvement in the performance when the augmentation policy is switched from horizontal flipping to RandAugment with Grayscale. Table 4 contains the test accuracy of different augmentation policies on 3-shot Pr $\to$ Ar and Rl $\to$ Pr domain adaption tasks of the Office-Home dataset with Resnet34.

**Effect of Noisy-Labeled Target Samples**: In SSDA, we have few labeled samples from the target domain; however, the presence of noisy-labeled target samples can have an adverse effect on the performance. To understand the effect of noisy-labeled target samples on the framework, we conducted experiments on the 1-shot Pr $\to$ Ar and Rl $\to$ Ar domain adaptation scenarios of the Office-Home dataset with Resnet34, where we mislabeled some previously labeled target samples as shown in Table 6. We observe a small decrease in performance of our framework ( from $66.2\%$ to $65.7\%$ for Pr $\to$ Ar and from $72.6\%$ to$71.56\%$ for Rl $\to$ Ar) when mislabeled target samples increase from $0\%$ to $\sim 25\%$ in both domain adaptation scenarios showing the robustness of our framework.

## 5 Conclusion

In this work, we present a novel single-stage contrastive learning framework for semi-supervised domain adaptation. The framework consists of Inter-Domain Contrastive Alignment and Instance-Contrastive Alignment, where the former maximizes the similarity between centroids of the same class from both domains and later maximizes the similarity between augmented views of the unlabeled target images. We show that both of the components of the framework are necessary for improved performance. We demonstrate the effectiveness of our approach on three standard domain adaptation benchmark datasets, outperforming the well-known SSDA methods.

## 6 Acknowledgments and Disclosure of Funding

The work is supported by Half-Time Research Assistantship (HTRA) grants from the Ministry of Education, India. We would also like to thank Saurav Chakraborty and Athira Nambiar for their valuable suggestions and feedback to improve the work.

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
