# CLDA: Contrastive Learning for Semi-Supervised Domain Adaptation ( Supplementary Material )

**Ankit Singh**
Department of Computer Science
Indian Institute of Technology, Madras
`singh.ankit@cse.iitm.ac.in`

## 1 Overview

The supplementary material consists of the following.

- Implementation details of the CLDA approach.
- Additional Results of the DomainNet dataset for 5 and 10-shot settings with Resnet34 as backbone network are shown in Table 1.
- Performance evaluation of the CLDA approach on 1-shot setting of the office-home dataset using both Alexnet and Resnet34 models. Results are reported in Tables 2 and 3
- Discussion on Limitations and Societal Impacts.

## 2 Implementation Detail

The architecture of the network is similar to [2]. All other hyperparameters used in our framework are described in the main paper. We perform all our experiments on *Nivida Titan X GPU*. We present the complete implementation of our approach in Algorithm 1. The reported results in the main paper are achieved through one-time training. Here, we provide the mean performance of our approach with standard deviation on the office-Home dataset for 3-shot domain adaptation tasks in Table 4 using Alexnet as the backbone model.

## 3 Performance Analysis with more shots

We additionally conducted experiments on 5-shot and 10-shot domain adaptation tasks of the DomainNet dataset with Resnet34. We used the data splits released by [1] for experimentation. We evaluated our approach on all the domain adaptation scenarios as described in [2]. Our approach achieves superior results on all domain adaptation tasks showing the effectiveness of our framework.

## 4 Results on Office-Home for 1-shot

We further provide the results for the 1-shot setting of the Office-Home dataset in Tables 2 and 3 using Alexnet and Resnet34 as backbone models, respectively.

## 5 Limitations and Societal Impacts

It is well known that deep neural networks face the problem of miscalibration, *i.e.*, they are overconfident about incorrect prediction, which may result in images being pushed into wrong clusters,

35th Conference on Neural Information Processing Systems (NeurIPS 2021).

which adversely affects the performance. Though Instance Contrastive Learning improves pseudo-label accuracy, other advances in clustering approaches should be explored. A potential direction of research is to develop better and efficient ways of mining confident pseudo labels.

The UDA and SSDA aim to transfer the knowledge from the source domain to the target domain. This knowledge transfer comes with the basic presumption that the source model is unbiased. Any knowledge transfer will propagate the inherent bias to the target domain if there is some bias in the source model. When such a model with its inherent bias gets deployed, it may cause disadvantages to certain people. Thus, ensuring the source model is not inherently biased before any knowledge transfer is vital for fair treatment.

---

**Algorithm 1:** CLDA - **C**ontrastive **L**earning for Semi-Supervised **D**omain **A**daptation

---

**Input:** Source dataset $\{\mathcal{D}_s\}$, Labeled Target dataset $\{\mathcal{D}_{lt}\}$, Unlabeled Target dataset $\{\mathcal{D}_t\}$, and Model $\{\mathcal{G}, \mathcal{F}\}$

1 **for** *steps* 1 *to totalsteps* **do**

2     Load a mini-batch of source samples $\{(\mathbf{x}_i^s, y_i^s)\}_{i=1}^{i=B}$ from source dataset $\mathcal{D}_s$ and target labeled samples $\{(\mathbf{x}_i^{lt}, y_i^{lt})\}_{i=1}^{i=B}$ from labeled target dataset $\mathcal{D}_{lt}$

3     Compute $\mathcal{L}_{sup}$ *cross-entropy loss* on both source and labeled target samples.

4     Load a mini-batch of unlabeled target samples $\{(\mathbf{x}_i^t\}_{i=1}^{i=\mu \times B}$ from target dataset

5     Compute $\mathcal{L}_{ins}$ *Instance Contrastive Alignment* on input and strongly augmented unlabeled input images.

6     Assign the class to the unlabeled target samples based on their pseudo-label.

7     Update source centroids

8     Compute $\mathcal{L}_{clu}$ *Inter-Domain Contrastive Alignment* between unlabeled target samples and source samples.

9     Update $\{\mathcal{G}, \mathcal{F}\}$ using total loss $\mathcal{L}_{tot} = \mathcal{L}_{sup} + \alpha * \mathcal{L}_{ins} + \beta * \mathcal{L}_{clu}$

---

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

| Net | Method | R to C | R to P | P to C | C to S | S to P | R to S | P to R | MEAN |
|---|---|---|---|---|---|---|---|---|---|
| | | | | | **Five-shot** | | | | |
| | S+T | 64.5 | 63.1 | 64.2 | 59.2 | 60.4 | 56.2 | 75.7 | 63.3 |
| | DANN | 63.7 | 62.9 | 60.5 | 55.0 | 59.5 | 55.8 | 72.6 | 61.4 |
| | CDAN | 68.0 | 65.0 | 65.5 | 58.0 | 62.8 | 58.4 | 74.8 | 64.6 |
| Resnet34 | ENT | 77.1 | 71.0 | 75.7 | 61.9 | 66.2 | 64.6 | 81.1 | 71.1 |
| | MME | 75.5 | 70.4 | 74.0 | 65.0 | 68.2 | 65.5 | 79.9 | 71.2 |
| | APE | 77.7 | 73.0 | 76.9 | 67.0 | 71.4 | 68.8 | 80.5 | 73.6 |
| | CLDA | **80.3** | **76.0** | **77.8** | **71.6** | **74.5** | **72.9** | **84.0** | **76.7** |
| | | | | | **Ten-shot** | | | | |
| | S+T | 68.5 | 66.4 | 69.2 | 64.8 | 64.2 | 60.7 | 77.3 | 67.3 |
| | DANN | 70.0 | 64.5 | 64.0 | 56.9 | 60.7 | 60.5 | 75.9 | 64.6 |
| | CDAN | 69.3 | 65.3 | 64.6 | 57.5 | 61.6 | 60.2 | 77.0 | 65.1 |
| Resnet34 | ENT | 79.0 | 72.9 | 78.0 | 68.9 | 68.4 | 68.1 | 82.6 | 74.0 |
| | MME | 77.1 | 71.9 | 76.3 | 67.0 | 69.7 | 67.8 | 81.2 | 73.0 |
| | APE | 79.8 | 75.1 | 78.9 | 70.5 | 73.6 | 70.8 | 82.9 | 76.8 |
| | CLDA | **81.2** | **77.7** | **80.3** | **74.1** | **77.1** | **74.1** | **85.1** | **78.5** |

Table 1: **Classification accuracy (%) on the DomainNet dataset with the Resnet34 backbone on 5-shot and 10-shot settings.** We have highlighted the best method for each domain adaptation task. Numbers show top-1 accuracy values for different domain adaptation scenarios. CLDA surpasses all the baseline methods in all adaptation scenarios.

| Net | Method | Rl→Cl | Rl→Pr | Rl→Ar | Pr→Rl | Pr→Cl | Pr→Ar | Ar→Pl | Ar→Cl | Ar→Rl | Cl→Rl | Cl→Ar | Cl→Pr | Mean |
|---|---|---|---|---|---|---|---|---|---|---|---|---|---|---|
| | | | | | | | **One-shot** | | | | | | | |
| | S+T | 37.5 | 63.1 | 44.8 | 54.3 | 31.7 | 31.5 | 48.8 | 31.1 | 53.3 | 48.5 | 33.9 | 50.8 | 44.1 |
| | DANN | 42.5 | 64.2 | 45.1 | 56.4 | 36.6 | 32.7 | 43.5 | 34.4 | 51.9 | 51.0 | 33.8 | 49.4 | 45.1 |
| | ADR | 37.8 | 63.5 | 45.4 | 53.5 | 32.5 | 32.2 | 49.5 | 31.8 | 53.4 | 49.7 | 34.2 | 50.4 | 44.5 |
| Alexnet | CDAN | 36.1 | 62.3 | 42.2 | 52.7 | 28.0 | 27.8 | 48.7 | 28.0 | 51.3 | 41.0 | 26.8 | 49.9 | 41.2 |
| | ENT | 26.8 | 65.8 | 45.8 | 56.3 | 23.5 | 21.9 | 47.4 | 22.1 | 53.4 | 30.8 | 18.1 | 53.6 | 38.8 |
| | MME | 42.0 | 69.6 | 48.3 | 58.7 | **37.8** | 34.9 | 52.5 | 36.4 | 57.0 | 54.1 | 39.5 | 59.1 | 49.2 |
| | BiAT | - | - | - | - | - | - | - | - | - | - | - | - | 49.6 |
| | CLDA | **45.0** | **72.6** | **51.5** | **62.4** | 37.1 | **40.0** | **61.4** | **37.2** | **61.5** | **59.4** | **43.2** | **61.3** | **52.7** |

Table 2: **Performance evaluation of Office-Home dataset on 1-shot setting using Alexnet.** Values show classification accuracy of different domain adaptation scenarios on 1-shot setting using Alexnet. Best results are marked in bold. CLDA surpasses all the baseline methods in most adaptation scenarios. Our Proposed framework achieves the best average performance among all compared methods.

| Net | Method | Rl→Cl | Rl→Pr | Rl→Ar | Pr→Rl | Pr→Cl | Pr→Ar | Ar→Pl | Ar→Cl | Ar→Rl | Cl→Rl | Cl→Ar | Cl→Pr | Mean |
|---|---|---|---|---|---|---|---|---|---|---|---|---|---|---|
| | | | | | | | **Three-shot** | | | | | | | |
| | S+T | 55.7 | 80.8 | 67.8 | 73.1 | 53.8 | 63.5 | 73.1 | 54.0 | 74.2 | 68.3 | 57.6 | 72.3 | 66.2 |
| | DANN | 57.3 | 75.5 | 65.2 | 69.2 | 51.8 | 56.6 | 68.3 | 54.7 | 73.8 | 67.1 | 55.1 | 67.5 | 63.5 |
| | ENT | 62.6 | 85.7 | 70.2 | 79.9 | 60.5 | 63.9 | 79.5 | 61.3 | 79.1 | 76.4 | 64.7 | 79.1 | 71.9 |
| Resnet34 | MME | 64.6 | 85.5 | 71.3 | 80.1 | 64.6 | 65.5 | 79.0 | 63.6 | 79.7 | 76.6 | 67.2 | 79.3 | 73.1 |
| | Meta-MME | 65.2 | - | - | - | 64.5 | 66.7 | - | 63.3 | - | - | 67.5 | - | - |
| | APE | **66.4** | 86.2 | 73.4 | 82.0 | **65.2** | 66.1 | 81.1 | **63.9** | 80.2 | 76.8 | 66.6 | 79.9 | 74.0 |
| | **CLDA** ( 1 shot) | 60.2 | 83.2 | 72.6 | 81.0 | 55.9 | 66.2 | 76.1 | 56.3 | 79.3 | 76.3 | 66.3 | 73.9 | 70.6 |
| | **CLDA** ( 3 shot) | 66.0 | **87.6** | **76.7** | **82.2** | 63.9 | **72.4** | **81.4** | 63.4 | **81.3** | **80.3** | **70.5** | **80.9** | **75.5** |

Table 3: **Results Analysis in Office-Home.** We perform experiments on all domain adaptation tasks of the Office-Home datasets using Resnet34 in both 1 and 3-shot settings. We have highlighted the best method for each transfer task. CLDA surpasses all the baseline methods in most adaptation scenarios. CLDA with only one labeled target sample per class achieves superior performance than **DANN** method with three labeled samples per class.

| Rl→Cl | Rl→Pr | Rl→Ar | Pr→Rl | Pr→Cl | Pr→Ar | Ar→Pl | Ar→Cl | Ar→Rl | Cl→Rl | Cl→Ar | Cl→Pr | Mean |
|---|---|---|---|---|---|---|---|---|---|---|---|---|
| $51.67 \pm 0.25$ | $74.33 \pm 0.35$ | $54.55 \pm 0.28$ | $66.84 \pm 0.24$ | $47.45 \pm 0.61$ | $44.77 \pm 0.38$ | $66.15 \pm 0.54$ | $47.20 \pm 0.29$ | $66.67 \pm 0.12$ | $64.32 \pm 0.37$ | $46.61 \pm 0.22$ | $67.16 \pm 0.42$ | $58.31 \pm 0.01$ |

Table 4: **Performance of multiple runs of CLDA on Office-Home in 3 shot setting using Alexnet.** We report the mean performance and its standard deviation for two runs of the CLDA approach on the Office-Home dataset in 3 shot setting. Standard deviation reflects the stability of our proposed method.