# OpenReview forum: "CLDA: Contrastive Learning for Semi-Supervised Domain Adaptation"
_NeurIPS.cc/2021/Conference — NeurIPS 2021 Poster_

### Official Review · Reviewer_bD47 · 2021-07-16

**Rating:** 6
**Confidence:** 4

**Summary:**

This paper proposes a contrastive learning method for semi-supervised domain adaptation (CLDA), where few labelled samples from target domain are available. The CLDA consists in inter-domain contrastive alignment, which decreases distance between same classes from source and target domains, and instance contrastive alignment based on strongly augmented unlabeled target images to reduce intra-domain discrepancy. The experiments are reported on three commonly used domain adaptation datasets.



**Limitations And Societal Impact:**

The limitation of the proposed method regarding inaccuracy of pweduo labels is not favorably addressed. See Main Review for my suggestion.

**Main Review:**

Strengths

1. The proposed method is simple and makes sense. It is interesting to see that the contrastive learning is effectively applied to semi-supervised domain adaptation problem.

2. The ablation studies are thorough and performance on three benchmarks are good.

Weaknesses

1. The inter-domain contrast alignment depends on the pseudo labels of the target samples. It is well known that some of the pseudo labels are inaccurate, which may be hurtful for the centroid estimation of target categories. Some recent studies [34,33, R1] have investigated different ways for mining high-confident pseudo labels and I am wondering how the proposed method will improve by considering this situation.

2. The instance contrastive alignment only makes use of target samples. It will  also be interesting to consider the contrastive loss for the source samples, particularly as a regularizing term for the supervised, cross-entropy loss.

3. Some recent works on semi-supervised domain adaptation are missing, e.g., [R1-R3]. It is necessary for the authors to cite these advances and compare with them. In particular, connection with and difference from R3 should be discussed and clarified.


[R1] Cross-Domain Adaptive Clustering for Semi-Supervised Domain Adaptation. CVPR, 2021.
[R2] Learning Invariant Representations and Risks for Semi-supervised Domain Adaptation. CVPR, 2021.
[R3] Prototypical Cross-domain Self-supervised Learning for Few-shot
Unsupervised Domain Adaptation. CVPR, 2021.

----------------------------------------------------------------
After rebuttal
The authors' responses basically addressed my concerns and I have also read the comments of other reviewers. I keep my initial rating and recommend acceptance of this paper.


**Time Spent Reviewing:**

10

---

> ### Author Response · Authors · 2021-08-10
> **Response to Reviewer bD47**
>
> We want to thank the reviewer for the feedback. It is encouraging to see that the reviewer find our work interesting.
>
> Our paper emphasized proposing a simple yet efficient contrastive learning framework for SSDA. Proposing different ways for mining high-confident pseudo-labels will be a manuscript on its own. We thank the reviewer for suggesting some recent work on mining confident pseudo-labels. We experimented by including [33] in our approach where weights of the unlabeled target samples used to estimate the centroid of a class are calculated using the entropy conditioning method explained in [33]. We obtained the below results for the 3-shot Pr $\rightarrow$ Ar and Rl$\rightarrow$Ar domain adaptation scenarios using Resnet34.
>
> |Approach| Pr $\rightarrow$ Ar | Rl $\rightarrow$ Ar|
> |------| ---------| ------|
> |With Entropy Conditioning| 72.6 | 76.4|
> |Without Entropy Conditioning| 72.4 | 76.7 |
>
>
>
> As per the review, We experimented on 3 shot Pr $\rightarrow$ Ar and Rl $\rightarrow$ Ar domain adaptation scenarios in the Office-Home (Resnet34) by employing contrastive loss on the source samples apart from the other losses described in the paper, but we did not get any improvements over the baseline. Results obtained are as follows:
>
> |Approach| Pr $\rightarrow$ Ar | Rl $\rightarrow$ Ar|
> |------| ---------| ------|
> |**With** Instance contrastive alignment on source samples| 71.7 | 75.3|
> |**Without** Instance contrastive alignment on source samples| 72.4 | 76.7|
>
>
> **Recent Works**
>
> We would like to thanks the reviewer for highlighting some recently published CVPR 2021 works. [R3] proposes a contrastive framework for unsupervised domain adaptation where very few source samples (3 % and 6% for office-home) are labeled while our paper deals with a different SSDA setting. The Contrastive framework proposed by [R3] aligns features and centroid within and across the domain, while CLDA proposes contrastive alignment both at the class and instance level for SSDA.
>
> Comparison with concurrent CVPR 2021  works (3 shot setting using Resnet 34).
>
> **Office-Home**
>
> |Approach|Rl$\rightarrow$Cl| Rl$\rightarrow$Pr |Rl$\rightarrow$Ar |Pr$\rightarrow$Rl |Pr$\rightarrow$Cl |Pr$\rightarrow$Ar| Ar$\rightarrow$Pl |Ar$\rightarrow$Cl| Ar$\rightarrow$Rl| Cl$\rightarrow$Rl| Cl$\rightarrow$Ar |Cl$\rightarrow$Pr |Avg|
> |---|---|---|---|---|---|---|---|---|---|---|---|---|---|
> |CDAC[R1]|67.8 |85.6 |72.2 |81.9 |67.0 |67.5| 80.3 |65.9 |80.6 |80.2 |67.4 |81.4 |74.2|
> |LIRR[R2] (5%) | | 83.45| ||62.84||||76.63|||||
> |CLDA (ours)|66.0 | 87.6 |76.7 |82.2 | 63.9 | 72.4 |81.4 |63.4 |81.3 |80.3 |70.5 |80.9 |**75.5** |
>
>
> **DomainNet**
>
> |Approach|R $\rightarrow$ C |R$\rightarrow$P|P$\rightarrow$C|C$\rightarrow$S|S$\rightarrow$P|R$\rightarrow$S|P$\rightarrow$R|Avg|
> |---|---|---|---|---|---|---|---|---|
> |CDAC [R1]|79.6 |75.1| 79.3 | 69.9 | 73.4 | 72.5 | 81.9 | **76.0**|
> |LIRR [R2] (5%)|63.03|||54.44|||||
> |CLDA (ours)|77.7|75.7|76.4 |69.7 | 73.7 |71.1 | 82.9 |75.3 |
>
> **Office31** (Alexnet)
>
> |Approach | W $\rightarrow$ A | D $\rightarrow$ A | Avg.|
> |---|---|---|---|
> |CDAC | 70.1 | 70 | 70.0|
> |CLDA (ours) | 70.5 | 72.5 | **71.5**|
>
> LIRR [R2] has a different setting than CDAC, CLDA. It also doesn't use the data splits released by MME [42]. We obtained superior results on the Office-Home and office31 dataset and competitive results on the DomainNet dataset showing our approach's efficacy against the concurrent CVPR works.

---

### Official Review · Reviewer_MSYR · 2021-07-16

**Rating:** 6
**Confidence:** 4

**Summary:**

The paper is about semi-supervised domain adaptation.  The inputs are labeled source samples, as well as labeled and unlabeled target domain samples.  The task is to train a model that performs well on the target domain.  The authors propose a framework with two main components to learn a domain-agnostic representation.

- Inter-domain contrastive alignment: This module reduces the discrepancy between classes centroids of the same class from source and target domain, while increasing the distance between class centroids of different classes from both source and target domains. Class centroids are computed as an exponential moving average via equations (2) and (3).  A contrastive loss is used to pull and push the centroids.
- Instance contrastive alignment: The goal of this module is to pull the classifier boundary into a low-density region, which is done by regularizing the classifier on the unlabeled target data.  Image augmentation is used to create perturbed images of unlabeled target data and a contrastive loss is employed to pull the representation of the same instance (perturbed and unperturbed) together while pushing it away from other samples.

Experiments on Office-Home, DomainNet and Office31 show promising results.

**Limitations And Societal Impact:**

- Limitations are okay, albeit being rather generic.
- Is there a possibility that the domain transfer/alignment method introduces additional biases, e.g., because the target data (although mostly unlabeled) is biased?

**Main Review:**

### Clarity

- It seems from lines 141ff that class centers for unlabeled target samples are computed via pseudo labels, which are then pushed closer to the class centers of the labeled data. But in the paragraph around Equation 4, I did not find anything about the labeled target images?! Shouldn't those also contribute to the class center and pushed towards the source domain class centers?

- Line 251ff: How is the instance contrastive alignment actually different to FixMatch?

- In Section 3.3, how do you back-propagate through the exponential moving average of the class centers. Doesn't this create a computational graph that spans over all previous training iterations?


### Experiments

- In line 193, the experimental setup is defined as to use the same **number** of labeled samples in the target domain. But are the exact same samples used as in prior work for the labeled target set or just the same number? I would assume the final performance (strongly) depends on **which** samples are chose; not all subsets of size N from the target domain dataset are equal!

- In Section 4.3, did you also compare with a sort of upper bound where all source and target data is labeled?

- Line 217: Does "3-shot setting" mean that 3 samples are labeled? I'm only asking because this nomenclature is often used for few-shot learning but not for semi-supervised learning.


### References

- There is a related work that I think could be discussed as related (concurrent) work [A].

- I think references to some relevant work on class-aware domain adaptation are missing, like [B,C]. These works use a class-specific discriminator in an adversarial domain adaptation scheme, where class information for unlabeled target data is also obtained via pseudo labels.

- In line 133, I think there could be a reference to "using the output space for adaptation": [D], although this paper does use an adversarial approach for domain adaptation.

- I think the sentence in lines 162ff ("This idea ...") should get a reference.

#### References
- [A] Domain Adaptation for Semantic Segmentation via Patch-Wise Contrastive Learning. Liu et al. arXiv'21
- [B] Ssf- dan: Separated semantic feature based domain adaptation network for semantic segmentation. Du et al. ICCV'19
- [C] Domain Adaptive Semantic Segmentation Using Weak Labels. Paul et al. ECCV'20
- [D] Learning to Adapt Structured Output Space for Semantic Segmentation, Tsai et al, CVPR'18



### Minor comments and typos

- Type in caption Figure 1: "... samples from both domains. (b) ..."
- Line 45: success ... success ?
- Line 53: "In summary, ..." & "... we propose ..."
- Line 90: "[24,26] use ..."
- Line 97: "... of limited labeled target ..."
- Line 118: The notation would be more consistent if the labeled source data is denoted as $\mathcal{D}_{ls}$
- Line 124: Remove the dot after $\mathcal{F}(\cdot)$
- Line 132: "[6,16] have ..."
- Line 159: "... is the strong augmentation function." More details are required at this point, e.g, a definition, a citation or a reference to the implementation details of the paper or something.
- Line 179: Use "eqref" to refer to equations.
- Line 183: "... Office-home [52] and Office31 [41]."
- Line 210: "... Maeta-MME [30] and ENT [14] ..."
- Line 240: "... accuracy drops from 72.4\% ..."
- Line 245: "... contrastive loss affects the ..."

### Post-rebuttal update

After having read all other reviews and the authors' responses, I have decided to keep my initial rating and recommend accepting the submission. I appreciate the authors' effort in providing all additional explanations and experiments.

**Time Spent Reviewing:**

4

---

> ### Author Response · Authors · 2021-08-10
> **Response to Reviewer MSYR**
>
> We thank the reviewer for the feedback. We will surely incorporate the missing references in our paper.
>
> **Aligning Label Target Samples**:  APE[27] has shown that labeled target samples are inclined with the source domain in the SSDA setting. Thus pushing very few label target samples along with unlabeled target samples towards the source domain does not add much gain.
>
> **Instance Contrastive Alignment**: FixMatch uses the pseudo-label from the weakly augmented images as the target for the strongly augmented image. While our Instance Contrastive Alignment tries to make the representation of the strongly augmented image similar to the weakly augmented image in a contrastive fashion.
>
> **BackPropogation**: CLDA tries to cluster the unlabeled target samples around the source prototype. We know that source prototypes are reliable due to the corresponding labels associated with source samples. So, backpropagation only takes place for unlabeled target samples.
>
> **Dataset**: We used the dataset splits realised by MME for all SSDA experiments. Thus, we are using the same number of samples and the same exact samples for the labeled target domain across all our experiments.
>
> **Comparison**: In section 4.3, **S+T** involves training with only labeled source samples and labeled target samples without using unlabeled target samples. Thus, it acts as a lower bound for comparison against the competing methods.
>
> 3-shot setting indicates that three samples per class of the target domain are labeled.

---

### Official Review · Reviewer_nEH7 · 2021-07-17

**Rating:** 6
**Confidence:** 4

**Summary:**

The authors propose a method for semi-supervised domain adaptation. In their methodology, contrastive prototypical alignment is used for the source and target examples (pseudo labels are computed for unlabeled target examples) and contrastive loss is used for the unlabeled examples from the target domain. The hope is that by using the contrastive loss on the unlabeled examples, the target examples would form clusters and the classification boundaries would lie on the low density data regions.

**Limitations And Societal Impact:**

Limitations are discussed.

**Main Review:**

Strengths:
-------------
- The paper is very well written. Easy to read and follow.
- Everything including datasets and implementation details are well described
- Experiments on multiple datasets and  ablation studies are also well conducted.
- They achieve improved performance in most experiments over the baselines.

Weakness:
--------------
- The novelty is very limited. The method proposed in 3.2 Inter domain contrastive alignment is very close to other protortypical alignment methods in unsupervised and semi-supervised domain adaptation.
- Its is not well understood how contrastive learning on unlabeled examples help. The authors say it clusters the target examples better and point to improved overall performance as evidence for it. We know that clusters formed by self-supervised learning methods do not necessarily correspond to classes of interest.  I do believe in this case, a theoretical justification and an empirical evaluation of the type of formed clusters is needed.


**Time Spent Reviewing:**

6

---

> ### Author Response · Authors · 2021-08-10
> **Response to Reviewer nEH7**
>
> We thank the reviewer for their thoughtful feedback. We are encouraged that they found our method clear and our analysis well described.
>
> **Novelty**: Our novelty lies in introducing the simple yet powerful contrasting learning-based framework for SSDA. Our paper proposes contrastive alignment at the class level and instance level to reduce the discrepancies in the SSDA  for superior performance gains. At the same time, the prior works only align the samples either at class level or instance level. In Section 3.3, we introduced a modified variant of Simclr[6] as Instance Contrastive Alignment. We also explored different settings for data augmentation, which is beneficial for contrastive alignment in SSDA.
>
> **Self-Supervised learning**: Usage of different self-supervised approaches in semi-supervised settings have been explored in [R1, R3] and has given superior performance over other SSL(semi-supervised learning) approaches. Self-supervised learning in a semi-supervised setting can lead to better representation, and it ensures samples are far from the classification boundaries and into a high-density region resulting in superior performance.
>
>
> -------------------------------------------------------------------------------------
>
> [R1]: S4L: Self-Supervised Semi-Supervised Learning, ICCV 2019
>
> [R2]: Self-Supervised Spatiotemporal Feature Learning via Video Rotation Prediction, Arxiv

---

### Official Review · Reviewer_aZTZ · 2021-07-18

**Rating:** 6
**Confidence:** 5

**Summary:**

This work presents a contrastive learning framework for Semi-Supervised Domain Adaptation. A basic assumption of few labeled target samples is made according to the Semi-Supervised setting. The key idea is to apply an instance-level and inter-domain contrastive alignment, using labeled source samples, few labeled target samples, and pseudo-labeled target samples. For instance-level contrastive loss, strong augmentations are used as positive pairs. For inter-domain alignment, cluster centroids are aligned across the two domains. The proposed method shows consistent improvement over multiple datasets.

**Limitations And Societal Impact:**

Adequately addressed.

**Main Review:**

Strengths
=========

a. The approach is simple. It is interesting to see that the two contrastive learning losses collaboratively improve the performance.

b. The results look promising.

c. The manuscript is easy to read


Weaknesses / Scope of improvement
==============================

a. Comparison against prior arts:

* The proposed contrastive losses are inspired by prior works [24, 26], however no comparison is made against these methods in the experiments (or did I miss it?).

* Since the proposed method uses strong augmentation, which can be seen as additional data, it is expected that the performance would improve. However, the results reported for the baselines are taken from their papers (L210). Considering that the baseline papers use weak augmentations (e.g. MME [42] uses horizontal flipping and random cropping), in my opinion, this comparison is unfair. The baselines should be evaluated using strong augmentations as shown in Table 4 (e.g., consider RA + Grayscale augmentation for competing methods).


b. Quality of pseudo-label / few-shot samples: According to Eq. 4, all the target samples are assigned a pseudo-label which is the model’s predicted output. This would also contain misclassified target samples. However, there is no discussion on how to mitigate the noise in pseudo labels. Moreover, there is no discussion on the performance when the one-shot labeled target samples are outliers.

c. L283 (“Instance Contrastive Learning improves pseudo-label accuracy”): It is unclear whether this is truly the case, since the individual modules don’t yield high performance on their own (L242). Moreover, there is no discussion regarding how the combination of the two losses gives significant improvement. It would be interesting to study why this is the case.

d. Supplementary L11-12 (“results in the main paper are achieved through one-time training”): Could the authors clarify if multi-run statistics are reported in the main paper? For instance, it seems that the result presented in Table 1 of the paper (Office-Home - Alexnet - CLDA - Mean = 57.4%), is around 1% less than that mentioned in the Supplementary Table 4 (Mean = 58.31%). Moreover, this discrepancy suggests that the results do not truly indicate the average-case performance of this method. The authors are suggested to present multi-run statistics for all experiments, since the performance depends upon the chosen one-shot / few-shot samples.


Additional clarifications
===================

a. Algorithm 1 (Supplementary): The source centroids are updated using the exponential average as shown in Eq. 3. However, the target centroids are not updated likewise. Could the authors clarify the intuition behind this choice?

b. How is the validation set constructed (Fig. 3) and used? Does this require labeled target samples beyond the few-shot labeled samples used for training? Is the validation set required for every experiment?

c. It is a bit surprising that such a simple method is able to outperform other methods by a significant margin. Could the authors confirm if the test set and the few-shot labeled samples are identical across all the compared methods?

d. L207 (“Complete details of all the experiments are included in the Appendix.”): Unfortunately, I could not find any relevant detail about the experiments in the Supplementary. Please update the supplementary to include details about the aforementioned points.


Minor comments
===============

a. Page 9: “Figure 4” -> “Table 4”

b. L53: “Summary” -> “summary”

c. Please maintain consistency in references (e.g. [1] is cited as NeurIPS while [4] cited as NIPS; [32] is published in ICML 2015, but cited as arXiv etc.).

d. A convergence plot could be useful in determining the stability of the training.


Considering both the strengths and weaknesses, I would suggest a revision at this stage. I would be happy to increase the score if my major concerns are addressed (see Weaknesses and Additional Clarifications).

--------

Post Rebuttal Comments: My major concerns were addressed in the authors' responses. Therefore, I am increasing the score.

**Time Spent Reviewing:**

6

---

> ### Author Response · Authors · 2021-08-10
> **Response to Reviewer aZTZ**
>
> We want to thank the reviewer for the feedback.
>
> **Comparison with Prior Works**
> [26] is more of a pretraining approach for domain adaptation and uses  MME [42] for domain alignment. Thus [26] is more of a complementary approach rather than a competing method. Also, [26] is proposed for UDA where the source domain labels are also limited, which is quite different from SSDA. CAN[24] deals with the UDA setting, so we took the official code of CAN[24] and introduced labeled target samples to make it work in the SSDA setting. However, the results obtained with CAN[24] were not encouraging; e.g., we obtained only  $59 %$ accuracy for Pr $\rightarrow$ Ar with Resnet34 on the Office-Home dataset in a 3-shot setting. So, we stick to the baselines used in the prior works.
>
> **Comparison of  MME [42] with strong augmentation on unlabeled samples with CLDA**:
> In our humble opinion, comparison with the baselines paper should be on the numbers reported by them. Any changes in the method should be presented as an ablation study rather than as the original approach.
> As per the review, we replaced the weak augmentation in the  MME [42] approach with strong augmentation for unlabeled target samples and obtained the below results on the Office-Home dataset using Resnet34.  MME [42] with strong augmentation shows poor performance on the Office-Home dataset. Strong Augmentation of images can result in poor performance in the absence of supervisory signals. We stick to the numbers reported by APE as it performs perturbation in the direction of the anisotropically high entropy of the target features.
>
> **Office-Home**
>
> |Approach|Rl$\rightarrow$Cl| Rl$\rightarrow$Pr |Rl$\rightarrow$Ar |Pr$\rightarrow$Rl |Pr$\rightarrow$Cl |Pr$\rightarrow$Ar| Ar$\rightarrow$Pl |Ar$\rightarrow$Cl| Ar$\rightarrow$Rl| Cl$\rightarrow$Rl| Cl$\rightarrow$Ar |Cl$\rightarrow$Pr |Avg|
> |---|---|---|---|---|---|---|---|---|---|---|---|---|---|
> | MME (with strong augmentation) | 58.86|83.10 |73.16 |77.18 |58.74 |67.02| 75.04 | 56.63| 75.08| 74.22|67.33 |75.35 |70.14 |
> | MME (with weak augmentation)|64.6| 85.5 |71.3| 80.1| 64.6 |65.5 |79.0 |63.6 |79.7 |76.6| 67.2| 79.3| 73.1 |
> |CLDA (ours)|66.0 | 87.6 |76.7 |82.2 | 63.9 | 72.4 |81.4 |63.4 |81.3 |80.3 |70.5 |80.9 |75.5 |
>
>
> **Quality of Pseudo-Label/ few-shot samples**:  Our paper empirically shows that a combination of inter-domain contrastive alignment and instance contrastive alignment improves the accuracy of domain alignment. However, our paper emphasizes on providing a simple yet effective strategy for SSDA and incorporating/proposing advanced methods for mining pseudo-labels for unlabeled target samples will create a manuscript of its own and could be a good idea for future research work.
> Also, since we do not explicitly use the labeled target samples, our method would be more robust to outliers or wrongly labeled samples in the target domain. To verify the claim, we experimented on 1 shot Pr $\rightarrow$ Ar and Rl $\rightarrow$ Ar domain adaptation scenarios using Resnet34, where we mislabeled some labeled target samples and computed the performance of our approach under these settings.
>
> | Mislabeled samples | Pr $\rightarrow$ Ar (Accuracy)| Rl $\rightarrow$ Ar(Accuracy) |
> |---|---|---|
> |8 samples mislabeled (approx 12 %) | 66.0  |72.05|
> |16 samples mislabeled (approx 25 %) | 65.7   |71.56|
> |without Misclassification| 66.2| 72.6|
>
> It clearly shows that our method is resilient to outliers in labeled target samples.
>
> **Instance Contrastive Learning improves pseudo-label accuracy**:  Instance Contrastive Alignment without Inter-Domain Contrastive Alignment ensures that unlabeled target samples are consistent and are in the high-density region. However, there is no alignment between source and unlabeled target samples. Inter-Domain Contrastive Alignment without Instance Contrastive Alignment reduces the discrepancy between unlabeled target samples and source domain but unlabeled target samples closer to the decision boundary might get pushed towards the wrong classes resulting in negative transfer. Thus combining both methods result in a much better alignment of the unlabeled target samples towards the source domain.
>
>
> **Results in the main paper are achieved through one-time training** We used the data splits released by  MME [42] for our experiments. Following  MME [42], we also reported results achieved through one-time training. However, we reported the multi-run statistics in the supplementary for Office-Home to show the stability of our approach. As suggested, we will indeed incorporate the average-case performance of all our experiments.
> The discrepancy between Table 1 of the paper (Office-Home - Alexnet - CLDA - Mean = 57.4%) and Supplementary Table 4 (Mean = 58.31%) is due to an error in jotting down the performance for one of the experiments Ar$\rightarrow$Rl, which we corrected in the supplementary. The original performance for Ar$\rightarrow$Rl domain adaptation scenario was $66.6 %$, which we wrongly jotted to $56.6 %$ in the main paper. We are incredibly sorry for this mistake.
>
>
> **Additional Clarifications**:
>
> a. we aim to align the unlabeled target samples around the source prototypes, so we ensure the alignment between unlabeled target samples with the EMA of the source samples. However, aligning source samples with the EMA of target samples can result in negative transfer due to the presence of noisy target samples in the calculation of the centroid of the target domain. To verify this, we experimented on 3 shot Pr $\rightarrow$ Ar and Rl $\rightarrow$Ar domain adaptations scenario using Resnet34 on the Office-Home dataset, where we also align source samples with EMA of target samples, and we obtained results in line with our intuition.
>
> |Approach| Pr $\rightarrow$ Ar | Rl $\rightarrow$ Ar|
> |---| ---| ---|
> |**With** Alignment of Source samples to EMA of target | 70.9 | 74.7|
> |**Without** Alignment of Source samples to EMA of target| 72.4 | 76.7|
>
> b. Validation splits for all the datasets were provided by  MME [42], and we have used the same in our experiments.
>
> c.Yes, the test set and few-shot labeled samples are consistent across all the competing methods.  MME [42] released the training, validation and test split for all the datasets, and the same was consistently used throughout all the methods and experiments.
>
> d.Though we tried to cover all aspects of the experiments either in the main paper or supplementary. We will surely consolidate experiment details across individual datasets and put them up separately in the supplementary.

---

> > ### Comment · Reviewer_aZTZ · 2021-08-27
> > **Thanks for the response**
> >
> > Thanks to the authors for the response. My major concerns are addressed; I would like to increase the score.
> > I would recommend the authors to incorporate the clarifications into the paper (especially, the discussion on the experimental setup and the noise resilience).

---

### Decision · Program_Chairs · 2021-09-27

**Decision:**

Accept (Poster)

**Comment:**


 Overall, the reviewers found the proposed method simple but interesting, the paper well-written, and the experiments/ablations thorough. The reviewers raised a number of concerns including comparisons to related works (and concurrent work) and fairness of comparisons [aZTZ, bD47, MSYR], pseudo-labeling noise [aZTZ, bD47], as well as smaller clarifications and analysis. The authors provided a thorough rebuttal with significant additional experiments, including even comparisons to the concurrent CVPR works. The reviewers have been overall satisfied with the responses and some have decided to increase their scores.

As a result, I recommend acceptance and strongly encourage the authors to incorporate the many experimental results and clarifications into the revised paper, including discussions on the experimental setup and noise resilience.